power and energy systems

fusion energy, model credibility, simulation, design, validation

**Author for correspondence:**
Richard J. Taylor
e-mail: richard.taylor-4@manchester.ac.uk

# An integrated digital framework for the design, build and operation of fusion power plants

Eann A. Patterson[1], Sally Purdie[2], Richard J. Taylor[3] and Chris Waldon[2]

[1]School of Engineering, University of Liverpool, Liverpool L69 3GH, UK
[2]Culham Centre for Fusion Energy, Abingdon OX14 3DB, UK
[3]Dalton Nuclear Institute, University of Manchester, Manchester M13 9PL, UK

EAP, 0000-0003-4397-2160; RJT, 0000-0001-8201-9677

The development of a commercial fusion power plant presents a unique set of challenges associated with the complexity of the systems, the integration of novel technologies, the likely diversity and distribution of the organizations involved, and the scale of resources required. These challenges are reviewed and compared to those for other complex engineering systems. A framework for creating a digital environment that integrates research, test, design and operational data is discussed and is based on combining the integrated nuclear digital environment (INDE), proposed recently for nuclear fission power plants, with the hierarchical pyramid of test and simulation used in the aerospace industry. The framework offers the opportunity to plan modelling strategies that allow large design domains to be explored prior to optimizing a detailed design for construction; and in this context, the relationship between measurements and predictions are explored. The use of the framework to guide the socio-technical activity associated with a distributed and collaborative design process is discussed together with its potential benefits and the technology gaps that need to be addressed in order to realize them. These benefits include shorter development times, reduced costs and improvements in credibility, operability, reliability and safety.

## 1. Introduction

The world needs large-scale, clean, reliable energy sources. Fusion is considered to be one of the technologies that can provide the answer as it has 'the highest energy density technology with the lowest conceivable footprint' [1]. Realizing fusion at an

affordable cost is an ambitious goal that is being pursued through international collaboration that requires tackling a wide range of significant scientific and engineering challenges. There are some similarities between the design of fusion and fission power plants, including dealing with the challenges of irradiation and the design of the balance of plant and heat recovery, so that some of the ideas proposed recently for an integrated nuclear digital environment (INDE) for nuclear fission power plants [2] are also relevant to fusion power plant. However, there are significant differences associated with the generation of the heat which justify the exploration of more sophisticated digital frameworks to support the design, build and operation of the complex, multi-physics system required to generate fusion energy.

Fusion is the process that powers the Sun and all other stars, where atomic nuclei collide together and release energy in the form of neutrons. To extract energy from fusion, gas from a combination of isotopes of hydrogen—deuterium and tritium—is heated to very high temperatures, of the order of 100 million degrees Celsius. One way to achieve these conditions is through 'magnetic confinement'—controlling the hot gas, known as a plasma, with strong magnets. The most promising device for this is the 'tokamak', a ring-shaped magnetic chamber. In addition to the extreme temperatures required, an ultra-high vacuum is also needed to sustain the plasma which makes the design of the fusion power plant very different to a fission power plant.

In simple terms, in a fusion reactor tritium and deuterium react together in the plasma to form helium and high-energy neutrons; the neutrons are absorbed in the reactor chamber lining or breeder blanket, which breeds tritium from lithium and converts neutron energy into thermal energy. The helium atoms produced in the reaction compromise the demanding conditions required, and impurities introduced from the chamber wall cause cooling of the plasma. Mechanisms to minimize impurity accumulation serve to promote high levels of plasma performance that in turn could support the generation of efficient electrical power.

These design principles are the focus of ITER, a large-scale fusion experiment being undertaken by 35 nations and hosted in France. The ITER experiment is designed to demonstrate controlled burning plasma physics regimes and the validity of key enabling technologies, such as the breeder blanket. The scientific programme is described in the ITER research plan [3]. The following step in developing fusion energy into a viable technology will be the DEMO plant which will aim to produce 500 MW of electrical output and to demonstrate the integrated operation of a fusion power plant, but is not planned to compete in the energy market [4]. The European development of fusion energy is described in the EUROfusion roadmap [5]. Similarly, Japan [6], and many of the ITER international partners, are making plans for their fusion power plant class devices. In the case of both ITER and DEMO, the conceptual design and the design optimization require extrapolation, from limited existing data and understanding, to operating regimes far beyond current experience, i.e. there is an almost complete lack of physical measurement data. They also involve complex systems with emergent behaviour for which there are no reliable predictive models, in part because the emergence results in a loss of causality. However, ITER must show compliance to strict nuclear regulations and DEMO will provide the first fusion energy to the grid, so appropriate approaches to design and validation must be found. The fusion environment is harsh and subjects internal reactor furniture to demanding and unique concurrent processes from high heat fluxes, irradiation from highly energetic neutrons, varying magnetic flux and plasma erosion. The interaction of fast neutrons with an atomic nucleus in a crystalline material can transfer enough energy that the atom can leave its lattice location. In addition to displacement damage, gas production inside the matrix of the material is also important together with property degradation, transmutation, embrittlement, swelling and irradiation creep. These circumstances challenge conventional engineering design methodologies, which are founded in regimes that integrate physical tests and computational models, over size scales from material coupons through components and sub-systems to full-scale machines. In the aerospace industry, this integration is often described by a pyramidal hierarchy [7]. A key aspect of the integrated approach to experiment and simulation is the use of quantitative validation methods [8,9] to establish credibility in predictions through demonstrating concordance [10] and quantifying uncertainty [11]. However, these approaches are difficult or unviable when design developments advance in large steps, i.e. transformative rather than evolutionary design, and in the presence of complexity that causes the behaviour of the coupled system to change with scale or to be different from the simple sum of its parts. The fusion process is subject to many size effects that make it difficult to produce representative sample and component tests, hence many elements of the design can only be validated by the building and operation of large-scale experiments, such as ITER. The design step from ITER to DEMO falls into this category and could be described as a product launch without any full physical

prototypes. A plasma physics step-ladder [12] approach offers a conceptual pathway to identify a power plant operating window but is strongly dependent on its relationship with reactor technologies. In a fusion power plant, the progression from design optimization to operational performance is characterized by the merging of validation of the complete system model with qualification of the power plant because validation of the model can only be performed using operational data. In other words, there is a gap between testing at the component level and the system level which generates a larger residual risk, when switching on the power plant for the first time, than is present with other types of system. Thus, a strategy is required to provide sufficient confidence for the large-scale investment without relevant evidence from an integrated physical prototype [13]. These issues have been recognized for some time, and efforts made to develop appropriate tools including a modular design approach [14], integrated material models [15] and a conceptual risk register [4]. The latter is particularly significant because the major uncertainties are epistemic and associated with the coupled engineering and physics of the fusion reactor, with the result that these uncertainties dominate the design and decision-making processes [16]. Although these efforts represent considerable progress, they are focused on individual aspects of the problem; and hence, in this study an attempt is made to provide a framework for managing the design–build–operate process in its entirety. This process usually involves international collaboration across disciplines, for instance the ITER breeder blanket is being designed and built by collaborating teams in China, Europe, Japan and Russia [17] (https://www.iter.org/newsline/264/1556). The aim is to offer a framework that will enable internationally dispersed teams of experts in different fields to engage in the process of long-term collaboration required to generate innovative and successful designs that possess the necessary credibility with stakeholders to attract the required investment.

## 2. Proposed framework

Nearly forty years ago, Harris [18] proposed a 'validated analytical hierarchy of models' for simulating complex battlefield scenarios which was developed by the Department of Defence in the USA into a pyramid depicting four levels of analysis, from the bottom up: engineering models; engagement models; mission level simulations; and theatre/campaign level simulations. The concept has been extended to the simulation of aircraft as complex systems at the apex of the pyramid with components, elements and coupons in the levels below [19]; and with each level divided into physical tests on one side and simulations on the other [7,20].

Recently, a conceptual framework has been proposed for an integrated nuclear digital environment (INDE) that extends from the prototype design of fission nuclear power plants though operations and decommissioning to storage and waste disposal [2]. A series of interconnected multi-scale, multi-physics computational models combined with digital measurement data from experiments, in-service monitoring and plant inspections are conceived as the digital environment. It is proposed that the digital environment could lead to shorter development times, reduced costs and increased credibility, operability, reliability and safety. It has been recognized that a number of technology gaps need to be closed before the complete framework can be implemented; however, the framework provides both a structure for technology collaborations and roadmap for research and development. The balance of plant is similar in both fission and fusion power plant, which provides some opportunity for transfer of technology and design; however, the delivery of the energy is substantially different. There is also a fundamental difference in design philosophy between the two concepts. In fusion, power plant costs will tend to accumulate through the design process merely to ensure a functioning reactor can be produced. This is largely due to the developmental nature of the technology and the absence of operational experience of full-scale power plant. With fission, power plant designs are more mature and operational experience is plentiful. In this case, costs will tend to accumulate through controlling known hazards revealed through operation and mitigating their consequences. This effect is exacerbated by the differences in passivity of the two systems particularly in a loss of power scenario. For this reason, the inclusion of cost and risk data within the fusion digital environment is particularly important in illustrating the commercial implications of decision making as a design develops.

Thus, to create a framework for the design–build–operate process of fusion power plants, the INDE framework developed for fission plants has been combined with the pyramidal hierarchy used in the aerospace industry to generate the schematic diagram shown in figure 1. The evolution of the interface between the real-world (blue shading in figure 1) and the virtual environment (orange

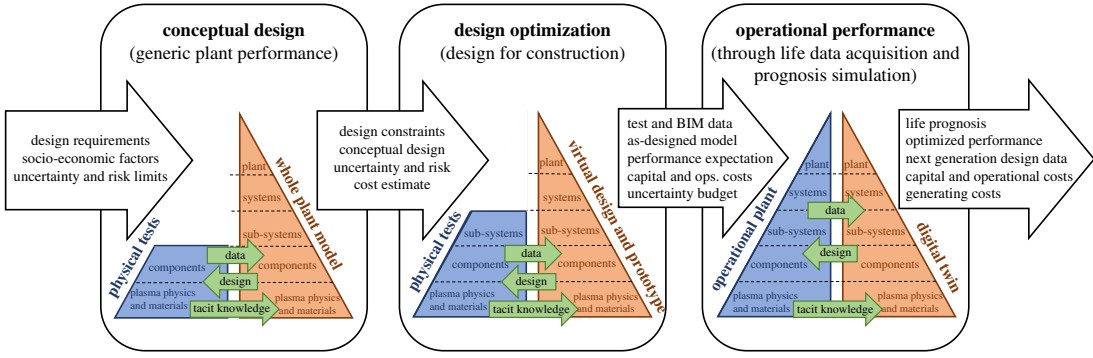

**Figure 1.** Schematic diagram of the evolving interface between the real-world (blue) and virtual environment (orange) as a fusion power plant progresses: from the conceptual design stage (left), when only limited physical test data is available; through the design embodiment stage (centre), including optimization of design with greater investment in tests and mock-ups; to an operational plant (right), when a digital twin is created from the accumulation of models and real-world data. At each stage, data from the real world supports the development of high-fidelity models through both realistic inputs and validation processes, while the models can be used to guide activity in the real world, as shown by the green arrows. The flow of information in and out of the stages are shown by the arrows at the top of the diagram.

shading in figure 1) is shown from the conceptual design stage when generic plant performance is explored, through optimization of the design culminating in design for construction, to an operational plant with an associated digital twin that can be used for simulation of performance and prognoses. The three horizontal stages in the schematic diagram are representative of the timeline of a design from concept to operation, such as illustrated qualitatively in the EUROfusion roadmap [5], and are equivalent to the virtual prototype and digital twin envisaged in the integrated digital environment for nuclear fission power plants. For fusion, the virtual prototype stage has been divided into conceptual design and design optimization to better represent the exploration of the wider design space that is needed.

The pyramid shown in each stage of figure 1 is based on the one used in the aerospace industry with the base extended to include plasma physics as well as materials science. At each stage, the virtual environment will consist of interconnected multi-scale and multi-physics models, including both principled models based on known theoretical knowledge and unprincipled models that might be parametric and based on empirical data. This series of models represents a whole power plant model at the conceptual stage that can be used to explore generic plant performance and becomes a digital twin when an operational power plant exists. The real-world side of the pyramid can only exist in its entirety once the power plant has been constructed and is operational, i.e. on the right of the schematic in figure 1. The nature of the fusion reaction means that, prior to construction, it is not possible to acquire real-world data for the power plant or its major systems and this causes the top portions of the real-world side of the pyramid to be missing at the design optimization stage; while even sub-system prototypes are unlikely to be manufactured at the conceptual design stage and so this portion is also missing in the first stage (on the left in figure 1). The lack of information from real-world measurements on sub-systems, systems and power plant is perhaps a unique feature of the design of fusion power plants and creates the need to extrapolate far beyond existing experimental data and technology to predict the behaviour and performance of an operational power plant. This lack of information drives a need to maximize the usage of the available physical data, as shown by the green 'data' arrows at each stage in the schematic diagram; but also, to optimize the design of physical experiments and tests using virtual models to maximize the information gained from the physical tests and experiments, as shown by the green 'design' arrows at each stage of the schematic diagram. The very long timeline represented by the schematic diagram in figure 1, which is typically decades, raises the imperative to capture tacit knowledge gained at each stage by embedding it in the virtual representation (represented by the third green arrow in each stage). Tacit knowledge includes the contextual and perceptual aspects of knowledge as well as the rationale underpinning design and modelling decisions [21].

While the flow of information between the real-world and the virtual environment, represented by the green arrows in figure 1, is essential to the integrated approach to experiment and simulation that is required to advance knowledge and understanding; the flow of information between the design stages is crucial in the development of a successful operating power plant. This latter process starts

with the identification of the design specification in terms of the operational requirements, the socio-economic factors and the acceptable limits of risk and uncertainty. These design attributes are shown as inputs to the process in the white arrow on the left of the schematic diagram in figure 1. In the concept design stage, a wide range of concepts drawn from a large design space should usually be considered, but only at a generic level to allow an assessment to be made about the degree to which they possess the design attributes, which for a complex system might involve multivariate analysis [22]. This will usually involve some physical tests to enhance understanding of key aspects of the underlying science and explore the behaviour of novel designs of engineering components; however, this physical testing is unlikely to involve the construction of any sub-system prototypes prior to the selection of a conceptual design for further embodiment and optimization in the next stage. The selection of the preferred conceptual design is likely to be made using a rational decision-making process [23] in which those designs lacking the essential attributes are rejected and the remaining designs are ranked on their possession of the desirable attributes. The design attributes, preferred conceptual design, and estimates of costs, risks and uncertainties are transferred from the conceptual design stage to the design optimization stage; and will take the form of both data from physical tests and experiments and from simulations, i.e. predictions. Much of this data will be ill-defined with large or undefined uncertainties including both aleatoric uncertainty, arising from inherent variability, and epistemic uncertainty due to a lack of knowledge about key parameters. In the design optimization stage, the embodiment of the design will justify the construction of physical sub-systems that will provide a higher level of test data than was available in the conceptual design stage. The design optimization will be focused in a single design concept that will be developed into a design for construction. This will require higher fidelity simulations of the design to provide data on performance expectation, capital and operational costs and an uncertainty budget; ultimately these data will be transferred into the digital twin of the operational power plant. The progression from conceptual design to design optimization should be marked by a decision on the preferred conceptual design but could still appear to be a somewhat fuzzy transition; whereas the progression from design optimization to operational performance will be very distinct because it will involve the construction of the power plant. Throughout the evolution of the power plant, building information modelling (BIM) data [24] should be acquired and added to the other inputs to the digital twin, as shown in the white arrow between the design optimization and operational performance stages in figure 1. During the operation of the power plant, physical data should be collected, including through health and conditional monitoring, plant inspections and financial audits, and transferred into databases associated with the digital twin. So that, the digital twin is a comprehensive representation of the power plant and its history, which will allow high-fidelity predictions of future costs, life prognoses and optimized performance to maximize operational metrics. There is a final flow of data from the operational performance into the design of the next generation of power plant, which is represented in more detail by the schematic diagram in figure 2.

The idealization in figure 1 is an over-simplification because it represents a single conceive, design, build and operate sequence with no subsequent development of the design; hence, in figure 2, simplified versions of the hierarchical pyramid from figure 1 have been used to construct a design cycle. This is a routine process in other industries, such as aerospace or automotive; however, in fusion energy it is less apparent because of the very long timescales involved. For instance, the construction phase for ITER started in 2007 and is scheduled for completion in 2025 (https://www.iter.org/construction/construction), which implies an 18-year timescale for the arrow between the design optimization and operational performance stages in figure 1. This implies that it is unlikely that any engineers or scientists will be involved in a complete design cycle of the type shown in figure 2; so that the proposed framework has a critical role in providing a structure for knowledge management.

It is intended to proceed to the DEMO plant using lessons learnt from ITER, and hence, it is relevant to discuss the ways in which the next generation process in figure 2 differs from the first generation one in figure 1. The long timescales mean that there is a high degree of concurrency, with fundamental science and engineering design being performed in parallel, and construction encapsulating the latest knowledge and understanding. This implies the need for a dynamic flow of information between stakeholders, which can be enabled by a shared and integrated digital framework.

The first three stages shown in figure 2 are identical to those shown in figure 1. However, step 4—next generation conceptual design differs from step 1—initial concept design, because there is less need to conduct physical tests, particularly at the scale of plasma physics and materials science, since this information is available from the physical tests conducted when developing the initial design and is

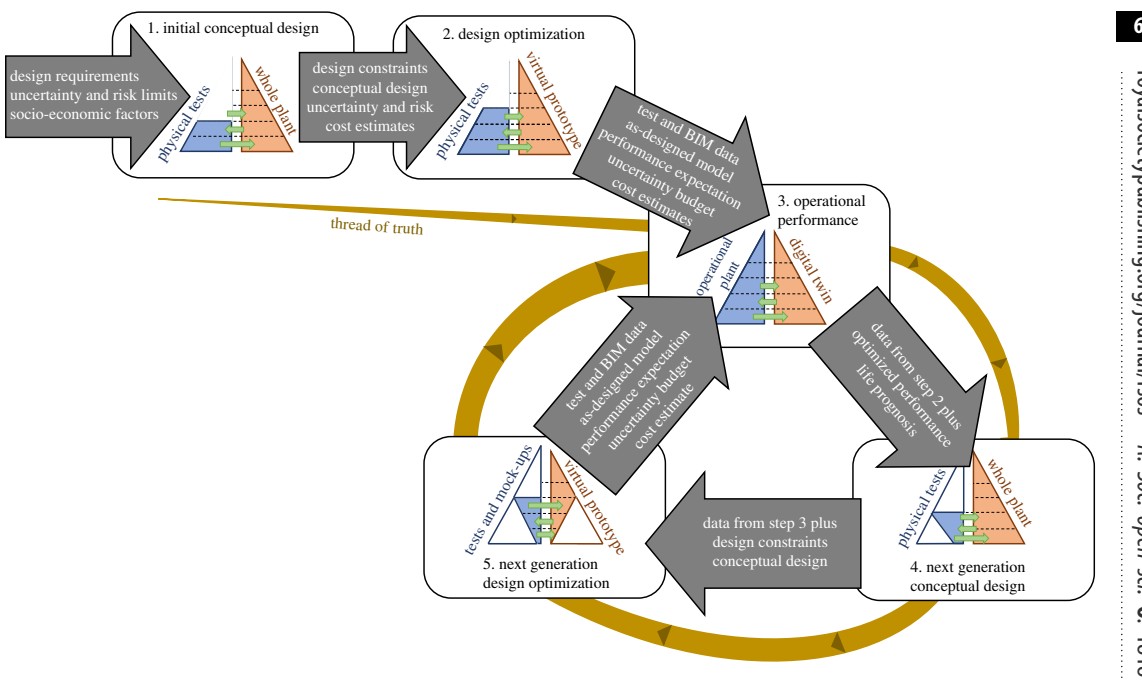

**Figure 2.** Flowchart illustrating the evolution of the test and modelling pyramid from the first generation of the concept, through a prototype to operational power plant and digital twin (from top left to centre right) as in figure 1; and subsequent cyclical evolution (clockwise around three stages on right) through successive generations, together with transfer of models and databases (grey arrows). The next generation conceptual design requires fewer physical tests due to the transfer of physical test and BIM data from the previous generation, as shown in bottom right pyramid; similarly, next generation design optimization benefits from historical physical test and operational data resulting in substantially less physical tests and mock-ups at plasma physics, components and sub-system levels; this is represented by the white portions of the pyramids. The 'thread of truth' is shown becoming larger as confidence in the simulations grows from stage to stage and from generation to generation.

supported by measurements and predictions from the operational power plant. The reduction in the need to conduct tests is represented by the smaller area of blue, while the existence of data is indicated by the unchanged blue boundary of the pyramid. The flow of information from the next generation conceptual design to the next generation design optimization stage is similar to that between the corresponding stages in the initial design, but with the addition of the previous generation data, preferably in the form of its digital twin. The previous generation digital twin will considerably reduce the need for predictive modelling in the optimization of the next generation design and this is represented by the white areas of the virtual prototype in stage 5 in figure 2. There are corresponding white areas in the physical test and mock-up side of the pyramid because previous generation data will be available and hence less testing will be necessary. Finally, the generational cycle is closed by the transfer of data from the construction and design processes to the new operational performance stage (stage 3 in figure 2); thus, allowing the generational cycle to commence again. The accumulation of historical design data and operational data through stages 1 to 5 will allow a progressively more comprehensive approach to assuring designs and validating models. This trend is illustrated by the increasing thickness of the 'thread of truth' in figure 2.

The diagram in figure 3 is based on a $2 \times 2$ matrix produced to illustrate the relationship between engineering models and physics together with approaches to establishing model reliability, i.e. fitness for purpose [25]. In the original $2 \times 2$ matrix, the horizontal axis was divided into known and unknown physics and the vertical axis into testable and untestable models. In figure 3, the divisions on the axes been replaced by continuous scales of knowledge and measured data strength respectively, so that a Boolean decision on the location in the plot of a model is unnecessary. The approaches to establishing model reliability are unchanged from the earlier work. In the bottom left corner, where measured data and knowledge strength are strong, a quantitative validation of model predictions can be performed following the standards and guides, e.g. ASME [8] and CEN [9], which is likely to result in a high level of reliability and credibility, as indicated by the dark grey shading. In the bottom right corner, only measured data strength is good implying that a model might be

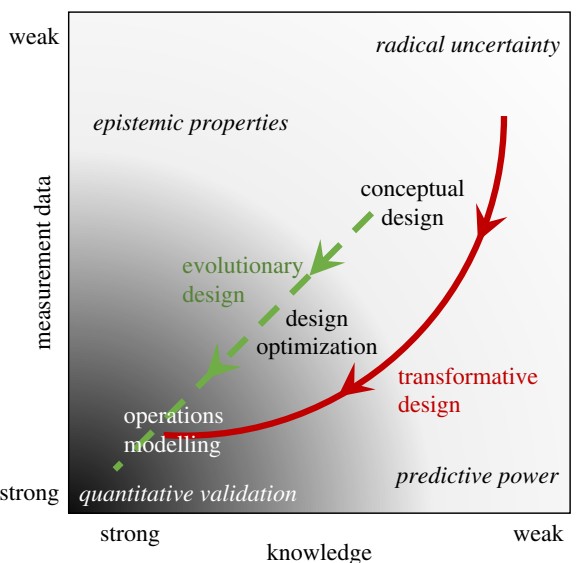

**Figure 3.** Schematic plot (based on Patterson [25]) showing two tracks of design development leading to increased credibility and reduced uncertainty in predictions of performance. Strength of knowledge decreases on the horizontal axis, and strength and availability of measurement data decreases on the vertical axis; so that models based on little or no data and knowledge in the top right represent radical uncertainty, while those based on detailed knowledge and data in the bottom left can be quantitatively validated or confirmed. The density of the grey shading indicates likely level of credibility that can be achieved; while the approaches to validation are shown in italics.

empirical without any foundation in scientific theory; in this situation Friedman's approach [26] to the validity of economic models can be adopted, i.e. the reliability of a model depends on its ability to predict rather than the veracity of its assumptions. The lack of knowledge will reduce the level of model credibility that can be achieved. In the top right corner, there is good knowledge strength and weak strength of measured data which means that quantitative comparison of measurements and predictions is not viable; instead, use must be made of the epistemic properties of the model [27], such as internal and external consistency, simplicity and explanatory power. Again, in this corner, only limited model credibility will be achievable. However, in the top right corner, it is unlikely that any model credibility can be achieved due to a lack of knowledge and data, which will result in radical uncertainty [28] and models that can only be used for heuristic purposes and not for making engineering decisions.

Most engineering design is evolutionary in form, in the sense that small or incremental changes are made from one generation of the design to the next generation, and those design changes that are unsuccessful are discarded. Consequently, they could be considered to start, as conceptual designs with limited knowledge and data describing their behaviour and performance, somewhere near the centre of the plot in figure 3 and to develop in a progressive way towards the bottom left corner with approximately equal increments in data and knowledge strength, providing support for the corresponding predictive models. This knowledge and data will be developed initially through the design optimization stage, with integrated experiments and simulation, and ultimately via operational performance. The result will be a substantial body of data and knowledge available to underpin a quantitative validation of predictive models of new generations of the design as they evolve. These circumstances apply to the balance of plant that is common to both fission and fusion power plants and hence allow the potential to 'plug and play' with well-established designs of these sub-systems. However, the start point in the conceptual design of the heat source in a fusion power plant is located towards the top right corner of figure 3 in the area of radical uncertainty because there is weak knowledge and measured data to support models coupling the engineering mechanics with the fusion process [4]. These models can be used to explore conceptual designs, with the possibility of substantial innovations leading to transformative design, and to support the design of physical tests and experiments that will provide data on which to base better models—this is the process represented by the conceptual design stage in figure 1 and the initial conceptual design stage in figure 2. The additional measured data moves the models down the plot in figure 3 allowing the prospect of increased credibility for the models. The complexity of the fusion reactor implies that our knowledge

and understanding of fusion power plants is likely to be acquired only after the accumulation of substantial quantities of data from operational plants, i.e. after achieving stage 3 in figure 2, at least for an initial generation and possibly for several generations, which implies that models of fusion power plants are likely to move to the left in figure 3 after the acquisition of data has moved them downwards. The translation of design models across the plot in figure 3 has implications for the approach taken to their validation or qualification and for the probability of achieving credibility among stakeholders.

## 3. Discussion

It has been suggested that the implementation of an integrated digital environment for nuclear fission would lead to the shortening of development times, reduced costs and improvements in credibility, operability, reliability and safety [2]. It is likely that these benefits would also accrue from the adoption of a digital or virtual framework for fusion power plants. For instance, the use of a shared digital environment between international collaborators and across the supply chain could lower the costs and risk associated with the construction of power plants, as is being achieved in production engineering [29]. A digital environment would also allow the design and rehearsal of maintenance procedures using virtual engineering technology, which would reduce the probability of issues arising during maintenance as well as the down-time required for maintenance [30]. This opportunity is particularly significant in fusion power plants as there will be an absence of repeat manufacturing and operational experience for the many bespoke or first of a kind (FOAK) components which would need to be used in a large-scale fusion reactor. A reduction in the risks associated with construction and maintenance is likely to encourage financial commitment from governments and eventually to lower finance costs for commercially funded power plants. Hence, in figures 1 and 2 it is proposed to go further than the integrated nuclear digital environment (INDE) proposed for nuclear fission power plants [2], by including modelling of capital and operational costs of fusion power plants [31,32] and estimates of the uncertainty associated with these predictions. This would allow the commercial returns from a fusion power plant to be predicted, taking account of construction and operational costs; and in turn, this is likely to reduce the price of capital funding of future fusion power plants through a lowering of the discount rate [33,34]. It is important to acknowledge that the realization of many of these benefits requires a level of accessibility that raises challenges for the protection of intellectual property, in both designs and modelling technology, and for the security of technology in terms of proprietary ownership and national concerns, for instance associated with export control regulations. It is likely that the solutions to similar issues in implementing INDE for nuclear fission will be transferable to fusion energy [2].

The implications of deploying a digital framework in the design of fusion power plants are perhaps more fundamental and far-reaching than in many other industries. This is due to the lack of operational plants, the limited physical test data, the lack of reliable predictive models of the coupled multi-physics processes [4], the need to extrapolate far beyond existing technology and data [4] and the drive to produce the first prototype power plant with an international collaboration. The latter creates a need for rapid and comprehensive access to design information by a team dispersed around the world in order to allow a coherent design to be developed on a reasonable timescale. In other industries, such as aerospace and automotive, design is undertaken by globally dispersed teams within single multi-national organizations and within trans-national supply chains using integrated digital environments based largely on commercially available software [35,36]. Often designs are transferred from one design office to another during a 24 h period as the working day finishes in one location and starts in another location, although this 'follow-the-sun' working practice is more common in software development [37]. In some cases, social media are used to capture and enhance the interactions of designers in a process termed social product development [38]. It is well known that effective and efficient communication and collaboration are among the most important success factors that influence productivity, lead-times and costs [39,40]. However, when the design, build, operate and decommission cycle is long, it is important to capture the tacit information embedded in the discussions within design teams so that future generations have access to the rationale underpinning the final design in order to support operation, maintenance and decommissioning decisions as well as next-generation designs. The framework proposed in figures 1 and 2 provides a structure within which the collaboration between dispersed design teams can take place and be captured.

The schematic in figure 1 is a combination of the linear evolution of a digital representation of the nuclear power industry proposed by Patterson *et al.* [2] and the pyramidal depiction of the

hierarchical integration of test and simulation used in the aerospace industry [7,19], while in figure 2, the concept is extended to describe its repeated application over a number of generations of design, build and operate. It is intended that the schematic in figure 2 should provide a framework for planning a pathway to the successful development of an operational fusion power plant taking account of the complexity of the machine, the long timescale of the processes and the multi-partner collaboration. Inevitably, the schematic in figure 2 contains a large degree of idealization; for instance, there will be a high level of concurrency between the acquisition of data from the physical and virtual implementation of ITER and the test and simulation of conceptual design features for DEMO, i.e. steps 3 and 4 in figure 2 will probably overlap with a fuzzy boundary between them. However, this probably strengthens, rather than weakens, the need for a framework within which activities can be planned and managed. For instance, the digital framework could be used to identify the hierarchy of problems to be tackled by defining the critical path and estimating the risks associated with each problem; so that the high-risk challenges on the critical path can be tackled first followed by the lower risk issues. Kemp *et al.* [4] have introduced a conceptual risk register than could be used in this process. This approach would also allow a shift away from the tendency to optimize single variables independently and towards consideration of trade-offs between several variables using, for example, multivariate analysis, which could lead to non-obvious solutions by lowering the cost of exploring such solutions [41].

At the moment, the probability of non-obvious solutions arising is reduced by the perceived requirements for validation of predictive models, as well as for design assurance [42] and product qualification [43]. There are no standards for validation, assurance or qualification processes associated with fusion power plants; and hence, fission standards are applied, which tends to substantially reduce the design space for fusion plants. Or, to view this relationship from the reverse perspective, the consideration of a large design space requires freedom from constraints, such as in the brain-storming phase of the rational decision-making process when the design attributes are set aside in order to generate a diverse set of design candidates [23]. The schematic in figure 3 implies that the expectations for validation must be relaxed at the stage of conceptual design, because the lack of knowledge and measurement data renders a quantitative validation impossible. There are some examples of this relaxation occurring, for example Maisonner *et al.* developed four conceptual designs for a tokamak that differ substantially in their physics, electrical output and blanket/divertor technologies; without any mention of validation or uncertainty quantification [44]. However, a systematic approach may be needed in which it is acknowledged that a lower level of fidelity is acceptable for exploring large design spaces [45] without the influence of designer prejudices or preconceptions while tracking the associated risks and uncertainties. These large design spaces could include those created by considering modular construction which might influence design against seismic events, or by three-dimensional printing which could lead to finer segmentation of the breeder blanket or to alternative replacement strategies for its surface following erosion by the plasma. The schematic in figure 3 provides a framework for planning the progression from conceptual design through design optimization to operations modelling. At the start of such a progression, knowledge and measured data is likely to be relatively unavailable, i.e. a model will be located in the top right of the schematic, and this implies that a quantitative validation process is probably not viable. Instead, the verisimilitude of predictions should be focused on establishing the epistemic properties of the model, since models with epistemic values are more likely to be appropriate than others [27]. Epistemic values include simplicity, explanatory power, internal consistency and external consistency. This approach has been discussed in a number of contexts include climate change modelling [27], where long-term predictions are sought but only short-term measurement data is available, and in computational biology [46], where it is often impossible to make measurements without disturbing the system of interest. When there is no measurement data available then the theoretical ancestry of the model needs to be identified and carried forward to provide some level of confidence in its predictions [47]. In addition, the credentials of the model builders and their techniques will also strongly influence the reliability of the predictions and should be similarly identified and recorded. Models are built on a series of assumptions that can be divided into tractability assumptions that render predictions viable and substantial assumptions that define the causal core of predictive models. The intention in developing a good model is that the tractability assumptions have no discernible influence on the predictions while the substantial assumptions lead to an acceptable representation of the real world. While it might be unviable to perform experiments with a fully integrated prototype, it is often possible to use reduced-scale experiments to confirm the veracity of assumptions [13]. The confirmation of assumptions will increase confidence and credibility in a model and its predictions but will not provide information about the fidelity of the predictions and the extent to which they are

an accurate representation of the real world, in other words this process falls short of being a validation process, as defined in the engineering guides [8,9]. However, models with verified assumptions but unvalidated predictions (located towards the top right in figure 3) are valuable for exploring large design domains where defining costs and timelines may be more important than other outputs, such as structural integrity. In this context, a lower level of engineering fidelity should be acceptable and establishing credibility becomes a higher priority. Yang *et al.* [48] proposed that establishing credibility in model predictions should be a continuous process involving interactions between stakeholders using a broad range of tools. They used focus groups and questionnaires, but the social media technology discussed above has important potential for both enabling and capturing these interactions.

The adoption of a digital or virtual framework for fusion power as outlined in figures 1 and 2 will necessitate overcoming particular challenges and the closure of a series of technology gaps if the potential advantages are to be realized. Given the prolonged timescales involved, the overall stewardship of the data held within the virtual framework will necessitate the development of new ways to capture and maintain both the explicit and tacit information [49], which has been recognized in some fields, for instance medicine [50]. An advance on current approaches to costing [51] will also need to be found to incorporate cost and schedule data into the framework in a manner that will allow integration with the technical content in real time. Construction of the digital twin will potentially bring together an unprecedented range of modelling platforms into a single space and a range of integrated codes will need to be developed to facilitate this integration in a manner that facilitates multiple users [52]. The key to success will be the establishment of best practice guidelines which describe the complexity/granularity of the digital environment appropriate to allow the framework to realize the advantages outlined above.

Additional challenges can be highlighted when considering the broader question of governance. A major infrastructure project such as a commercial fusion reactor is likely to experience changes of ownership or commercial accountability over its project life cycle. To maintain the integrity of the digital framework it will be necessary to exercise independent governance of the digital twin to ensure appropriate diligence at 'handover' or potentially even separate ownership as a national asset outside the commercial interests of the contractors delivering the project. Having recognized that designs for the commercial fusion plant are being developed to meet national requirements around the world, then this governance question will need to be considered on a country-by-country basis. Governance will need to include mechanisms for acceptance, assurance and consolidation of large quantities of design data prior to incorporation in the digital environment, as well as practical controls over data security. Compliance with industry or national standards, such as BIM or Industry 4.0, would also present an ongoing challenge.

A digital framework must incorporate large quantities of both design and power plant data; for this to be effective, a new generation of sensors/technologies will need to be developed which are capable of automatic model updates from real-time operational measurements. These developments will need to include both technologies specific to fusion reactors, such as monitoring of power output via neutron yields [53] and of surface morphology of plasma-facing components [54], e.g. breeder blankets, but also more conventional structural health monitoring during operations and non-destructive evaluation during manufacture and maintenance, for instance ultrasound examination of welds [55] and hydrostatic heat sinks [56,57].

If the above challenges can be overcome, then the use of a digital framework opens up new opportunities to harness the power of virtual and augmented reality, particularly in the field of promoting stakeholder confidence, and as an aid to conceptual and detailed design [58,59] that goes far beyond the limitations of a conventional three-dimensional CAD model and where it is already being implemented, for instance in automating construction [60] and in planning maintenance of fusion system components [61–63]. However, further research is required on quantifying the position of a system model in figure 3 and on communicating to stakeholders the implications of designing with models of low or ill-defined fidelity, such as those towards the top right in figure 3, particularly with regard to building confidence among stakeholders in the financial community. It is likely that any commercial fusion reactor will be delivered through a dispersed international collaborative design activity in some form of a 'social network', in which case it will be necessary to capture and archive the discussions and rationale underpinning design decisions. While some organizations have tailored social media tools to support their engineering activities [38,64], further work will be required to handle the multi-scale and multi-physics nature of a fusion power plant as well as the long timescales from its conception to decommissioning. In addition to the current design processes for ITER and DEMO being dispersed and international, the supply chain for a commercial fusion power plant is

likely to be both dispersed and diverse, i.e. in terms of size, structure and sector. Although there have been massive changes in supply chain management and the use of knowledge management, there remain some significant research issues including with the systems required to support knowledge management [65]. The digital environment based on the proposed framework has the potential to support the supply chain by providing access to design specifications and information about the supply chain, as well as acting as repository for information about equipment qualification and the provenance of components and sub-systems, for instance using digital ledgers [66]. However, a substantial effort will be required to generate a suitable information management structure that is secure but accessible to appropriate organizations, will remain functional over many decades and connects technology and business models [67]. These requirements probably imply that custodianship of the digital environment should belong to an international public body that is independent of commercial interests. It is worth noting that custodianship of the data of the digital environment is a different function to that of the design authority; and that data access is important for the earlier stages of the design process because it would allow innovative organizations to accelerate the design process by exploring non-obvious solutions and potentially to generate disruptive innovations [68,69]. The increasing social and political pressures to achieve zero-carbon economies are likely to become major drivers for the more rapid development of appropriate energy sources, including fusion; and hence, the enabling of disruptive innovations will probably increase in importance.

The approach to exploring conceptual designs using credible models with verified assumptions but unvalidated predictions is relevant to large-scale engineering endeavours for which the underlying physics is poorly understood and/or the complexity of the system is likely to create emergent behaviour that cannot be captured in reduced-scale experiments. Besides fusion energy, hypersonic flight, commercial space flight and the development of engines for interplanetary travel are examples of such endeavours where modelling with reduced levels of quantitative validation is likely to enable the exploration of large design spaces. All of these examples involve multi-disciplinary design teams working over long timescales and often dispersed across several organizations; so that the schematic diagrams presented here should provide a useful framework for managing these ambitious high-cost ventures and for providing access to information and knowledge in a form that is likely to provide opportunities for disruptive innovation.

## 4. Conclusion

A digital framework that integrates the many facets of the design, build and operation of future commercial fusion power plants is proposed. The framework needs to handle the issues associated with the computational representation of the multi-scale and multi-physics phenomena, including the emergent behaviour of the resultant complex system in both the physical and socio-economic domains. The issues connected to the distributed and collaborative design and construction processes need to be addressed over very long timescales and these reveal a number of technology gaps that need to be addressed, including:

(i)   methodologies for transformative design commencing from low-fidelity computational models, including communication of confidence to stakeholders;
(ii)  procedures for defining appropriate levels of granularity in digital twins;
(iii) mechanisms for incorporating costing and scheduling data in engineering models, and associated uncertainties;
(iv)  capacity to capture and maintain tacit knowledge over long timescales, including via social networking tools;
(v)   mechanisms for acceptance, assurance and consolidation of data into a digital twin of a fusion power plant, including the development of sensors and associated technologies capable of automated integration of measurements as well as data about the provenance of components and sub-systems.

It is proposed that independent stewardship of the resultant digital twin will be necessary to provide immunity from changes in commercial accountability over the life cycle of a power plant and to permit access for organizations likely to generate disruptive innovations.

The advantages of implementing an integrated digital framework for a fusion power plant include: the opportunity to explore a large design space potentially resulting in a more innovative design; the prospect of exploiting synergies with fission for balance of plant allowing the focus to be on

the specific features required for fusion; and shorter development and construction times resulting from better integration and sharing for knowledge and understanding of the design, which should also increase safety, reliability and operating standards. The philosophical approach implied by the framework is likely to increase stakeholder confidence, which could lead to reductions in finance costs and changes in the approach to regulatory approval.

Data accessibility. No data were used in the preparation of this paper.
Authors' contributions. E.A.P. and R.J.T. conceived the framework together following extensive discussions with S.P. and C.W. E.A.P. wrote the first draft of the manuscript. All authors provided feedback, extensively edited the manuscript and gave final approval for publication.
Competing interests. The authors have no competing interests.
Funding. No funding was received for this study.

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
