## [Reviewer comments · Royal Society Open Science]

Review History

RSOS-181847.R0 (Original submission)

Review form: Reviewer 1 (John William Stairmand)

Is the manuscript scientifically sound in its present form?

Yes

Are the interpretations and conclusions justified by the results?

Yes

Is the language acceptable?

Yes

Is it clear how to access all supporting data?

Not Applicable

Do you have any ethical concerns with this paper?

No

Have you any concerns about statistical analyses in this paper?

No

Recommendation?

Accept with minor revision (please list in comments)

Comments to the Author(s)

The paper provides a readable and informative account of how the digital twin approach could be applied to nuclear fusion technology. I thought there were several relevant additional issues that could have been mentioned, and which the authors should consider for inclusion. These are: 1) Many applications of the digital twin approach in non-nuclear industries relate to products that comprise multiple repeat manufacture. For nuclear fission, most new-build programmes have not involved this scale-out approach. For nuclear fusion, we will be in FOAK territory for a significant period. Nuclear fusion is therefore different in type from most conventional digital twin approaches.

2) Fusion technology involves many issues that take science and engineering beyond previous capabilities. Specifically, this includes high temperature, high vacuum, high heat flux, high neutron flux and high (and non-steady) magnetic flux. Separately these constitute engineering and modelling challenges. In combination, the challenge is even greater. The approach indicated in figure explains how this can be accommodated. However, for the benefit of readers not familiar with ITER, JET or DEMO, I suggest it is worth spelling out the challenge in more detail.

3) The concept of international open access working is laudable in principle. However, the authors should acknowledge the following obstacles:

3.1 Intellectual property protection (both of the design and of the codes)

3.2 Security considerations....for example Export Controls regulations

3.3 Quality Assurance, particularly relating to ascertaining verisimilitude of the output from a simulation in which at least one of the codes is likely to be based on empirical correlations, possibly being applied out of its validated domain,

Review form: Reviewer 2

Is the manuscript scientifically sound in its present form?

Yes

Are the interpretations and conclusions justified by the results?

Yes

Is the language acceptable?

Yes

Is it clear how to access all supporting data?

Not Applicable

Do you have any ethical concerns with this paper?

No

Have you any concerns about statistical analyses in this paper?

No

Recommendation?

Accept with minor revision (please list in comments)

Comments to the Author(s)

This is an interesting article transferring the knowledge and principles from nuclear fission and other areas towards the nuclear fusion research and power plant design. The described paths towards an optimised reactor design is well described and the different phases and cycles of optimisation proper explained.

However, I feel that not enough attention was paid to a few nuclear fusion peculiarities. The authors identified that ITER is the key device for the next step forward and that it provides vital input for subsequent devices. ITER is an international endeavor and therefore in my view only comparable to projects like the ISS or CERN. I stress this because the political fraction in the economic aspect is here in particular very high indeed. It might even rise due to the fact that nuclear fusion is a CO₂-free energy which might be the strongest driver in future. This aspect is not really reflected and I believe that it must.

ITER is a unique device in the moment. I advise to stress this fact of an international project in the paper stronger and adapt the reference as suggested. ITER – and there should be a quote to the recently research plan – will demonstrate or not the physics principles and already power amplifications close to a reactor. Thus, the step from ITER to a reactor is by far not like described a complete unknown territory. It might even be that a reactor will be of same size but optimized in performance and magnetic field strength. Thus, I advise to change slightly the tone when describing the path towards an unknown territory.

The authors then refer to an EUROPEAN DEMO design. In this case one shall stress that there is NOT one DEMO, but the different ITER partners themselves are developing devices which will fulfill the requirements of a DEMO as they themselves define. There is a Korean ITER design, a Chinese one based on CFETR, an EUROPEAN design (via EUROfusion PPPT), as well as Japanese and an American studies etc. The article suggests there is only ONE design, but the independent design of each country will likely permit different routes and reduce the risks. I recommend to clarify that the authors refer to an EUROPEAN design and that other designs are discussed at the same time in connection with the other partners.

I'm surprised to see no reference to the very comprehensive EUROPEAN DEMO research program which takes place under EUROfusion. There is even a roadmap published about the EUROPEAN way forward to an electrical power plant. The plan and structure includes comparable aspects and steps towards a design discussed here. Systems codes are used to predict the performance and give design boundaries for a reactor. They are more than the "whole plant model", but rather reflect "virtual designs and prototypes". An optimization on basis of the systems codes is already ongoing.

It is of course not the exact combination of the INDE and hierarchical pyramid as introduced here, but nevertheless it would be worth to indicate how the current plan and structure towards an optimized design looks like.

However, I feel that not enough attention was paid to a few nuclear fusion peculiarities

Decision letter (RSOS-181847.R0)

18-Jun-2019

Dear Professor Taylor,

The editors assigned to your paper ("An integrated digital framework for the design, build and operation of fusion power plants") have now received comments from reviewers. We would like

you to revise your paper in accordance with the referee and Associate Editor suggestions which can be found below (not including confidential reports to the Editor). Please note this decision does not guarantee eventual acceptance.

Please submit a copy of your revised paper before 11-Jul-2019. Please note that the revision deadline will expire at 00.00am on this date. If we do not hear from you within this time then it will be assumed that the paper has been withdrawn. In exceptional circumstances, extensions may be possible if agreed with the Editorial Office in advance. We do not allow multiple rounds of revision so we urge you to make every effort to fully address all of the comments at this stage. If deemed necessary by the Editors, your manuscript will be sent back to one or more of the original reviewers for assessment. If the original reviewers are not available, we may invite new reviewers.

- Data accessibility

<http://datadryad.org/submit?journalID=RSOS&manu=RSOS-181847>

- Competing interests

- Authors' contributions

- Acknowledgements

- Funding statement

on behalf of Professor Jin Jiang (Associate Editor) and R. Kerry Rowe (Subject Editor)
openscience@royalsociety.org

Associate Editor's comments (Professor Jin Jiang):

We apologise for the lengthy time your review has been under external assessment: the journal has had to approach an unusually large number of referees to secure the two reports we now have - the Editors express their gratitude to the reviewers for providing these comments, too. As you'll see, each reviewer is broadly positively inclined towards your review article. However, as each provides points for inclusion to improve the manuscript, we'd like you to revise the paper to address these concerns and ensure that you provide a point-by-point response to the suggestions/queries. The reviewers may be asked to confirm that you've addressed their points, so do make sure your changes are clearly indicated.

Thanks again for your interesting submission, and we'll look forward to receiving the revision in due course.

Comments to Author:

Reviewers' Comments to Author:

Reviewer: 1

Comments to the Author(s)

The paper provides a readable and informative account of how the digital twin approach could be applied to nuclear fusion technology. I thought there were several relevant additional issues that could have been mentioned, and which the authors should consider for inclusion. These are:

- 1) Many applications of the digital twin approach in non-nuclear industries relate to products that comprise multiple repeat manufacture. For nuclear fission, most new-build programmes have not involved this scale-out approach. For nuclear fusion, we will be in FOAK territory for a significant period. Nuclear fusion is therefore different in type from most conventional digital twin approaches.

- 2) Fusion technology involves many issues that take science and engineering beyond previous capabilities. Specifically, this includes high temperature, high vacuum, high heat flux, high neutron flux and high (and non-steady) magnetic flux. Separately these constitute engineering and modelling challenges. In combination, the challenge is even greater. The approach indicated in figure explains how this can be accommodated. However, for the benefit of readers not familiar with ITER, JET or DEMO, I suggest it is worth spelling out the challenge in more detail.

- 3) The concept of international open access working is laudable in principle. However, the authors should acknowledge the following obstacles:

- 3.1 Intellectual property protection (both of the design and of the codes)

- 3.2 Security considerations....for example Export Controls regulations

- 3.3 Quality Assurance, particularly relating to ascertaining verisimilitude of the output from a simulation in which at least one of the codes is likely to be based on empirical correlations, possibly being applied out of its validated domain,

Reviewer: 2

Comments to the Author(s)

This is an interesting article transferring the knowledge and principles from nuclear fission and other areas towards the nuclear fusion research and power plant design. The described paths towards an optimised reactor design is well described and the different phases and cycles of optimisation proper explained.

However, I feel that not enough attention was paid to a few nuclear fusion peculiarities. The authors identified that ITER is the key device for the next step forward and that it provides vital input for subsequent devices. ITER is an international endeavor and therefore in my view only comparable to projects like the ISS or CERN. I stress this because the political fraction in the economic aspect is here in particular very high indeed. It might even rise due to the fact that nuclear fusion is a CO₂-free energy which might be the strongest driver in future. This aspect is not really reflected and I believe that it must.

ITER is an unique device in the moment. I advise to stress this fact of an international project in the paper stronger and adapt the reference as suggested. ITER – and there should be a quote to the recently research plan – will demonstrate or not the physics principles and already power amplifications close to a reactor. Thus, the step from ITER to a reactor is by far not like described a complete unknown territory. It might even be that a reactor will be of same size but optimized

in performance and magnetic field strength. Thus, I advise to change slightly the tone when describing the path towards an unknown territory.

The authors then refer to an EUROPEAN DEMO design. In this case one shall stress that there is NOT one DEMO, but the different ITER partners themselves are developing devices which will fulfill the requirements of a DEMO as they themselves define. There is an Korean ITER design, an Chinese one based on CFETR, an EUROPEAN design (via EUROfusion PPPT), as well as Japanese and an American studies etc. The article suggests there is only ONE design, but the independent design of each country will likely permit different routes and reduce the risks. I recommend to clarify that the authors refer to an EUROPEAN design and that other designs are discussed at the same time in connection with the other partners.

I'm surprised to see no reference to the very comprehensive EUROPEAN DEMO research program which takes place under EUROfusion. There is even a roadmap published about the EUROPEAN way forward to an electrical power plant. The plan and structure includes comparable aspects and steps towards a design discussed here. Systems codes are used to predict the performance and give design boundaries for a reactor. They are more than the "whole plant model", but rather reflect "virtual designs and prototypes". An optimization on basis of the systems codes is already ongoing.

It is of course not the exact combination of the INDE and hierarchical pyramid as introduced here, but nevertheless it would be worth to indicate how the current plan and structure towards an optimized design looks like.

However, I feel that not enough attention was paid to a few nuclear fusion peculiarities

Author's Response to Decision Letter for (RSOS-181847.R0)

See Appendix A.

RSOS-181847.R1 (Revision)

Review form: Reviewer 1 (John William Stairmand)

Is the manuscript scientifically sound in its present form?

Yes

Are the interpretations and conclusions justified by the results?

Yes

Is the language acceptable?

Yes

Do you have any ethical concerns with this paper?

No

Have you any concerns about statistical analyses in this paper?

No

Recommendation?

Accept as is

Comments to the Author(s)

Thank you for including my comments in your revised submission. I am satisfied that the paper is now suitable for publication without further changes.

Review form: Reviewer 2

Is the manuscript scientifically sound in its present form?

Yes

Are the interpretations and conclusions justified by the results?

Yes

Is the language acceptable?

Yes

Do you have any ethical concerns with this paper?

No

Have you any concerns about statistical analyses in this paper?

No

Recommendation?

Accept as is

Comments to the Author(s)

The paper is fine in my view. An adequate response to the questions and issues which I have raised has been given.

Decision letter (RSOS-181847.R1)

08-Sep-2019

Dear Professor Taylor,

I am pleased to inform you that your manuscript entitled "An integrated digital framework for the design, build and operation of fusion power plants" is now accepted for publication in Royal Society Open Science.

You can expect to receive a proof of your article in the near future. Please contact the editorial office (openscience_proofs@royalsociety.org and openscience@royalsociety.org) to let us know if you are likely to be away from e-mail contact -- if you are going to be away, please nominate a co-

author (if available) to manage the proofing process, and ensure they are copied into your email to the journal.

on behalf of Professor Jin Jiang (Associate Editor) and R. Kerry Rowe (Subject Editor)
openscience@royalsociety.org

Reviewer comments to Author:

Reviewer: 1

Comments to the Author(s)

Thank you for including my comments in your revised submission. I am satisfied that the paper is now suitable for publication without further changes.

Reviewer: 2

Comments to the Author(s)

The paper is fine in my view. An adequate response to the questions and issues which I have raised has been given.

Follow Royal Society Publishing on Twitter: [@RSocPublishing](https://twitter.com/RSocPublishing)

Appendix A

RSOS-181847

An integrated digital framework for the design, build and operation of fusion power plants

Authors' responses to reviewers' comments

Reviewer: 1

The paper provides a readable and informative account of how the digital twin approach could be applied to nuclear fusion technology. I thought there were several relevant additional issues that could have been mentioned, and which the authors should consider for inclusion.

- Thank you for these positive comments. We have noted the additional issues that you raise and modified the text accordingly and highlighted the changes.

These are:

1) Many applications of the digital twin approach in non-nuclear industries relate to products that comprise multiple repeat manufacture. For nuclear fission, most new-build programmes have not involved this scale-out approach. For nuclear fusion, we will be in FOAK territory for a significant period. Nuclear fusion is therefore different in type from most conventional digital twin approaches.

- Yes, we acknowledge that we have not included this issue specifically. We have added further text in the discussion section that recognises the absence of repeat manufacturing experience for the many FOAK bespoke components within a large-scale fusion reactor. We have noted the particular challenge this presents in terms of component assurance and ongoing monitoring during operations.

2) Fusion technology involves many issues that take science and engineering beyond previous capabilities. Specifically, this includes high temperature, high vacuum, high heat flux, high neutron flux and high (and non-steady) magnetic flux. Separately these constitute engineering and modelling challenges. In combination, the challenge is even greater. The approach indicated in figure explains how this can be accommodated. However, for the benefit of readers not familiar with ITER, JET or DEMO, I suggest it is worth spelling out the challenge in more detail.

- Agreed. Text has been added to the introduction to make specific that it is these factors which account for the unique challenges of sub-system or prototypical scale testing of fusion systems.

3) The concept of international open access working is laudable in principle. However, the authors should acknowledge the following obstacles:

3.1 Intellectual property protection (both of the design and of the codes)

3.2 Security considerations....for example Export Controls regulations

3.3 Quality Assurance, particularly relating to ascertaining verisimilitude of the output from a simulation in which at least one of the codes is likely to be based on empirical correlations, possibly being applied out of its validated domain,

- The authors have attempted to recognise both the challenges and opportunities presented by allowing all stakeholders a level of 'open access' to the digital twin and the necessity for a new governance model for the twin which transcends short term commercial interests. We

have now added additional text to the discussion section that recognises the specific challenges described above.

Reviewer: 2

This is an interesting article transferring the knowledge and principles from nuclear fission and other areas towards the nuclear fusion research and power plant design. The described paths towards an optimised reactor design is well described and the different phases and cycles of optimisation proper explained.

- Thank you for these positive comments.

However, I feel that not enough attention was paid to a few nuclear fusion peculiarities. The authors identified that ITER is the key device for the next step forward and that it provides vital input for subsequent devices. ITER is an international endeavor and therefore in my view only comparable to projects like the ISS or CERN. I stress this because the political fraction in the economic aspect is here in particular very high indeed. It might even rise due to the fact that nuclear fusion is a CO₂-free energy which might be the strongest driver in future. This aspect is not really reflected and I believe that it must.

- We have inserted some additional details related to combination of special condition associated with current designs of fusion reactors including high temperature, high vacuum, high heat flux, high neutron flux and high (and non-steady) magnetic flux. In addition, we have included comment about the likely increase in political pressure associated with achieving zero-carbon economies.

ITER is an unique device in the moment. I advise to stress this fact of an international project in the paper stronger and adapt the reference as suggested. ITER – and there should be a quote to the recently research plan – will demonstrate or not the physics principles and already power amplifications close to a reactor. Thus, the step from ITER to a reactor is by far not like described a complete unknown territory. It might even be that a reactor will be of same size but optimized in performance and magnetic field strength. Thus, I advise to change slightly the tone when describing the path towards an unknown territory.

- The authors have revised the text to place more emphasis on the unique nature of the ITER plant and the ambition to demonstrate the necessary parameters to conceive, design and construct the DEMO facilities. A reference has been included to the ITER research plan and its targets.

The authors then refer to an EUROPEAN DEMO design. In this case one shall stress that there is NOT one DEMO, but the different ITER partners themselves are developing devices which will fulfill the requirements of a DEMO as they themselves define. There is an Korean ITER design, an chinese one based on CFETR, an EUROPEAN design (via EUROfusion PPPT), as well as Japanese and an American studies etc. The article suggests there is only ONE design, but the independent design of each country will likely permit different routes and reduce the risks. I recommend to clarify that the authors refer to an EUROPEAN design and that other designs are discussed at the same time in connection with the other partners.

- The authors have revised the text to recognise the range of DEMO concepts currently under consideration. This has also now been picked up under challenges in that a range of digital twins will be required that incorporate data from ITER but reflect the specific requirements of each countries DEMO plans.

I'm surprised to see no reference to the very comprehensive EUROPEAN DEMO research program which takes place under EUROfusion. There is even a roadmap published about the EUROPEAN way forward to an electrical power plant. The plan and structure includes comparable aspects and steps towards a design discussed here. Systems codes are used to predict the performance and give design boundaries for a reactor. They are more than the "whole plant model", but rather reflect "virtual designs and prototypes". An optimization on basis of the systems codes is already ongoing.

It is of course not the exact combination of the INDE and hierarchical pyramid as introduced here, but nevertheless it would be worth to indicate how the current plan and structure towards an optimized design looks like.

- The authors hope that the integrated digital framework might be adopted as a basis for further developing the EUROfusion roadmap. It was an omission on our part not to reference this document and this has been corrected.

However, I feel that not enough attention was paid to a few nuclear fusion peculiarities

- We have looked to further emphasise the unique nature of the fusion challenge through our responses to all the reviewers' comments.